# Assessment of Motor Abilities and Physical Fitness in Youth in the Context of Talent Identification—OSF Test

**DOI:** 10.3390/ijerph192114303

**Published:** 2022-11-01

**Authors:** Joanna Baj-Korpak, Marian Jan Stelmach, Kamil Zaworski, Piotr Lichograj, Marek Wochna

**Affiliations:** 1Department of Health Sciences, John Paul II University of Applied Sciences in Biala Podlaska, 21-500 Biala Podlaska, Poland; 2Department of Technical Sciences, John Paul II University of Applied Sciences in Biala Podlaska, 21-500 Biala Podlaska, Poland; 3Polish Athletic Association, 01-839 Warsaw, Poland

**Keywords:** talent identification, youth sports, testing, skill level, OSF test

## Abstract

(1) Background: Physical fitness during childhood is an important indicator of current and future health status. Defining physical fitness levels is a key element of talent identification in the training of children and adolescents. It is also crucial in developing a sports career path. This study sought to validate a physical fitness test (OSF test) and to determine fitness norms for children and adolescents with a special focus on talent identification within its particular sub-tests of endurance, speed, strength and power. (2) Methods: A total of 27,187 athletes who participated in the ‘Athletics for all’ (AFA) programme were included in the analysis. Physical fitness was assessed using a validated OSF test (3 × 10 shuttle run, standing broad jump, 1 kg medicine ball overhead throw, 4-min run). (3) Results: Four key motor abilities (speed, power, strength and endurance) were assessed in children and adolescents. The OSF test was normalised, i.e., a reference frame in the form of centile charts was developed. (4) Conclusions: The centile charts developed for particular parts of the OSF test make it possible to interpret scores in particular motor sub-tests and, first and foremost, enable users to compare a given score with results obtained by their peers.

## 1. Introduction

Physical fitness (PF) as an ability of physiological systems of the body to cooperate effectively makes it possible to perform activities of daily living with the least effort possible and simultaneously constitutes a prerequisite to staying healthy. A fit person is capable of performing work at school, doing household chores and still has enough energy to enjoy sports and other leisure activities [1,2]. At the same time, taking up moderate-to-vigorous physical activity (MVPA) increases PF and may contribute to limiting sedentary behaviours and improving positive health measures [3,4,5,6].

According to Caspersen et al. [7], PF is a set of attributes that are either health- or skill-related. Howley and Franks [8] define PF as a state of well-being with a low risk of premature health problems and energy to participate in a variety of physical activities.

PF is also understood as a general ability to perform activities of daily living safely [9]. Its levels depend on complex genetic factors and physical activity. It involves a number of specific components that kinesiology experts are interested in [10].

An athlete’s high level of PF is a key element of effectiveness in sports competition. If coaches know their athletes’ PF levels, they can select proper training loads, i.e., types of exercises as well as their quantity and quality. Moreover, regular measurements of PF levels can help to assess training effects [11].

Motor abilities are individual psychophysical properties that determine the level of movement capabilities [12]. They constitute a group of aptitudes conditioned genetically and shaped by environmental determinants [13]. Motor abilities are most often determined based on strength, speed, endurance, agility and power [14]. They are often interrelated with one another and with other determinants of performance in a given sport [15].

According to Ortega et al. [16], PF is a summary measure of the body’s capacity to take up physical activity, and it is also an important marker of health. During childhood it is considered to be a marker of current and future health status. There is ample evidence that points to a correlation between sedentary behaviours and health problems in children and adolescents. Unfortunately, sedentary behaviours are still on the increase [17]. The result is children’s obesity, which has become a serious global epidemic that causes social issues and puts a lot of strain on healthcare systems all over the world. It is proven that in order to control and prevent obesity in children, it is necessary to implement physical activity programmes [18]. Moreover, the findings of the latest reviews show that physically active adolescents manifest not only higher PA levels but also better body composition in all parameters related to body fat mass, which produces numerous health benefits in adult life [19,20]. Therefore, there is a need for a simple but reliable test assessing particular components of PF that could be carried out, inter alia, at schools during PE lessons.

The use of different tests is an indispensable form of examining and monitoring motor abilities [15]. The aim of such tests is to assess motor abilities in a simple and accessible manner. Each test used for assessing fitness should be reliable and valid [14]. However, to interpret results of such a test properly, it is necessary to have accurately calculated reference norms [21]. Therefore, different batteries of tests assessing selected components of PF related to health, sports skills or aptitudes of adolescents were developed [22,23].

In the previous decade, several studies regarding PF of children and adolescents were presented in the form of centile reference values for various tests; however, no curves were developed. The majority of these studies are linked to batteries of tests that were used in research projects. Some of them were implemented as national systems of supervising PF in school-age children, e.g., FITescola in Portugal [24], SLOfit in Slovenia [25], NETFIT in Hungary [26] as well as ALPHA [27], HELENA [28] or EUROFIT [29].

This study sought to validate a physical fitness test (OSF test) and to determine fitness norms for children and adolescents with a special focus on talent identification within its particular tests of endurance, speed, strength and power. An attempt was made to produce standard percentile values regarding PF of children and adolescents measured with the OSF test, taking into account sex and age. The centile charts developed will enable their users to identify PF levels in children and adolescents quickly and accurately. The OSF test employed in our study is a new tool for assessing PF. It covers four leading motor abilities, i.e., speed, power, strength and endurance. These are key motor abilities that indicate which groups of athletics events (sprints, jumps, throws or long-distance runs) one should get engaged in. We believe that the findings of our study will be used by coaches and teachers.

## 2. Materials and Methods

### 2.1. Participants

In total, 27,187 individuals (participants of the ‘Athletics for All’ programme) were included in the study.

The ‘Athletics for All’ (AFA) programme was created in 2014 by the Polish Athletic Association in order to promote athletics as ‘the first choice’ sport among children and adolescents. The aim of the programme is to show athletics as a versatile sport that makes it possible to derive a lot of joy and satisfaction from physical activity, competition with peers and from gaining new skills and experience. Currently, the programme includes over 600 training groups from all over Poland. Young athletes are trained by experienced athletic coaches and instructors.

The main aim of the programme is to select the most gifted children from thousands of participants. In the future, such children will undergo sport-specific training following general and performance-oriented programme. Training within AFA is based on certain stages. It takes into account participants’ age and level of advancement. General training is performed in the following age groups: stage 1–5–10 years of age (grades 1 to 4 of primary school), stage 2–10–15 years of age (grades 5 to 8 of primary school) [30].

Performance-oriented training (stage 3) is held in Centres of Oriented Training (COTs) with adolescents aged 14–19 [31].

In the youngest group, 90-min training sessions are held twice a week. Children from grades 5 to 8 take part in 90-min sessions three times a week, while in COTs 90-min sessions are performed at least five times a week. Stage 3 of the programme is the most advanced, and it is oriented at specific athletic blocks of events (matching athletes’ abilities). Thanks to the funding by the Polish Ministry of Sport and Tourism, the involvement of local governments as well as sponsors’ support, participation in the programme is free.

One of the main goals of AFA is to develop a national system of diagnosis, selection and identification of talented children within youth training. Another goal is to develop an athletics career path that would form the basis for a new structure of training organisation for children and adolescents in Poland. The current study corresponds with the guidelines provided by the Polish Athletic Association.

Mean age of the participants was 11.68 (±1.59) years and it was slightly lower than median age (12 years). Girls constituted a larger proportion of the population under study (57.8%)—see Table 1.

### 2.2. Methods

Physical fitness was assessed using the OSF test, which includes four validated sub-tests:3 × 10 m shuttle run—speed test-In a standing start position, the participant begins their run from pole 1 (start line) to pole 2 (finish line). The athlete runs around pole 2 counter-clockwise (with their left shoulder closer to the pole) and runs towards pole 1. The athlete runs around it in the same manner and heads towards the finish line. Each participant performs one warm-up run at a moderate pace and two measured trials. Measurements are performed with an accuracy of 0.01 sec.Standing broad jump—power test-With feet apart, the participant stands just behind a line marked on the ground (they cannot stand on the line). A two-feet take-off and landing is performed. The jump can be preceded by swinging of the arms. The athlete can lean forward and backward but they cannot take their feet off the ground. The distance is measured from the take-off line to the first contact point in the landing area. If the participant falls on the back, they can repeat their attempt. Apart from a warm-up trial, the athlete performs this test twice (one after another). A better result is noted down (with an accuracy of 1 cm).1 kg medicine ball overhead throw—strength test-With feet apart and facing the direction in which the ball is to be thrown, the athlete stands behind a line marked on the ground (they cannot stand on the line). The ball is held over the head with two hands. The ball is brought back behind the head, and then it is thrown forward without taking the feet off the round. After performing the throw, the athlete cannot cross the line. The distance is measured from the line to where the ball lands. The participant performs one warm-up trial followed by two measured trials (one after another). A better result is recorded (with an accuracy of 5 cm).4-min run—endurance test-The test can be performed by 4 to 6 individuals simultaneously. The athlete runs around the 10 m x 19.25 m rectangle. The rectangle is formed using four poles. Three markers are used to indicate its length (two markers situated 4.68 m from the poles, and a middle marker positioned 5 m from the two markers). One marker is used to indicate the width of the rectangle. The marker is located in the middle (5 m from the poles). The actual trial is preceded by a demonstration during which the participants run one lap around the rectangle. They stand behind the start line. At a signal (a whistle), they run counter-clockwise (with their left shoulder closer to the poles). During the run, a supervisor informs them of the time left: ‘3 min left, 2 min left, 1 min left, 30 s left, 15 s left, 10 s left, five-four-three-two-one’. At a signal (a whistle or ‘stop’), the participants finish their run. In the course of the trial, the supervisor notes down the number of laps completed. Afterwards, the number of laps is multiplied by the length of one lap, and the distance beyond the last lap is added. The result is recorded with an accuracy of 5 m. In the course of the test, the participants are allowed to walk if they feel very tired (it does not disqualify them).

Results of particular PF tests were converted into points (on a scale from 1 to 100) taking into account age and sex of the participants (Appendix A).

The OSF test reliability was checked using the findings of the pilot study carried out in 2014 on a sample of 30 primary school students. The sample size was determined in accordance with COSMIN recommendations [32]. The test was conducted twice (test-retest) on the same sample with a 3-week interval (21 days). The Cronbach’s alpha coefficient was used to measure internal consistency of the results (ICC). Minimum acceptable consistency was set at ≥0.75. It was revealed that the OSF battery of tests demonstrated high reliability for all four tests (ICC ≥ 0.90) [33].

The tool meets the reliability criteria, i.e., results are the same if the test is performed in the same conditions. The test was normalised, i.e., a reference frame in the form of centile charts was developed. The frame enables test users to make comparisons with the whole population in terms of a given feature.

Anthropometric measurements were made with the use of calibrated equipment. Each measurement was carried out twice under the same conditions.

Body height was measured with an accuracy of 1 mm using SECA 213 stadiometer, while body mass was registered using SECA 875 scales in accuracy class 3 (200 g). If the difference between the first and the second measurement was 300 g or more for body mass and 5 mm or more for body height, the third measurement was performed.

### 2.3. Organisation and Conduct of the Study

The study was carried out in the years 2015–2018 according to the schedule prepared in compliance with strictly defined rules (following OSF performance instructions). Prior to the study, all AFA coaches had been trained in terms of the study protocol. Each study participant provided a written informed consent signed by their parents or legal guardians to take part in the OSF test and to have their body height and body mass measured. A written consent to use the data collected in the course of the study for scientific purposes was also obtained from the Polish Athletic Association.

The Bioethics Committee of Warsaw University of Life Sciences, Faculty of Human Nutrition and Consumer Sciences (Resolution No. 16/2017) approved the protocol in accordance with the Declaration of Helsinki.

### 2.4. Statistical Analysis

Quantitative variables were described in terms of the parametric distribution (checked with Shapiro–Wilk test and Kolmogorov–Smirnov test) taking into account such descriptive characteristics as mean, median, standard deviation (SD), values of quartiles 1 and 3 as well as the range. For categorical data, frequency distribution of particular replies is presented using the size of particular categories and their distribution expressed in percentage terms.

The following tests were employed in the study: U Mann–Whitney test, Kruskal–Wallis test (with Dunn’s post hoc and Bonferroni correction for multiple comparisons), chi-square test and Fisher test. U Mann–Whitney test is a non-parametric test used for comparing distributions of numerical variables between two groups under observation. Statistically significant results obtained with it point to the occurrence of differences in the distribution of a given variable between these groups. This test is an alternative to the Student t test in case its assumptions (normality of distributions and homogeneity of variance) are not met. Kruskal–Wallis test is a non-parametric test also used for comparing the distribution of a given variable between more than two groups. In order to investigate the correlation between categorical variables, chi-square test or Fisher test was applied. Effect sizes, which are quantitative measures of force of a given phenomenon, were estimated using η2 effect coefficient. To check whether there are any monotonic correlations between variables, Spearman’s rank correlation coefficient was employed, which is a measure of monotonic statistical correlations between variables under study. This coefficient can range from −1 to 1. If its value is positive, it means that when the value of one variable increases, the value of the other one also rises. If the correlation is negative, it indicates that when the value of one variable increases, the value of the other one decreases. The strength of correlations can be classified as follows:0.0 < |r| ≤ 0.2—no correlation0.2 < |r| ≤ 0.4—weak correlation0.4 < |r| ≤ 0.7—moderate correlation0.7 < |r| ≤ 0.9–strong correlation0.9 < |r| < 1.0—very strong correlation.

For each sub-test, the distribution of results was visualised by drawing centile curves based on LMS function from GAMLSS package [34] in RStudio. The construction of centile curves involved one explanatory variable, i.e., the results of each of the four sub-tests of the OSF test taking into account sex. The model employed assumes that the explanatory variable has the following distribution: y ~ D (μ, σ, ν, τ), where smooth functions of the explanatory variable are parameters of this distribution, i.e., g (μ)= s (x), where g () is the so-called link function, while s () is a certain smooth function. LMS () function uses the method of spline functions to smooth them. The function proceeds to matching several proper distributions with the explanatory variable. A set of distributions to be matched is defined by groups of arguments, while a class of LMS distributions constitutes conjectural arguments, in accordance with the work of Cole and Green [35]: Box–Cox Cole Green original, Box Cox Power Exponential original and Box-Cox T original. The best model was selected through minimising global deviance (GD). In this manner, in all figures (from bottom to top), curves were drawn for the following centiles: 0.38, 2.27, 9.12, 25.25, 50, 74.75, 90.88, 97.72, 99.62, in accordance with the work of Cole [36].

Statistical significance was set at *p* ≤ 0.05. Significant results were also revealed for *p* ≤ 0.01 and *p* ≤ 0.001.

R statistical package (version 4.0.2) (Institute for Statistics and Mathematics, Wirtschaftsuniversität Wien, Vienna, Austria) was used to make all calculations and prepare charts. For each participant, normalised values of scores obtained in particular fitness tests were calculated. These values (*z scores*) were calculated according to the following formula:$$z\;\mathrm{score}\;=\;\frac{x_{k}-\bar{x}}{S_{x}}$$
where $x_{k}$*-* score value in a given test achieved by an athlete, $\bar{x}$—arithmetic mean of scores achieved by all athletes in a given fitness test, $S_{x}$*-* standard deviation of scores achieved by all athletes in a given fitness test.

Afterwards, for all *z scores* in all four OSF sub-tests, a total score was calculated using the following formula:$$\mathrm{Sum}=z\;score_{3x10m}+\;z\;score_{standing\;broad\;jump}+z\;score_{medicine\;ball\;throw}+z\;score_{4-minute\;run}$$

Due to the fact that a different scale was used for results obtained in 3 × 10 m run (lower scores were considered better), the sum was presented as a negative value.

Based on results achieved by all the participants, percentile values were produced. Furthermore, norm charts (percentiles of *z scores*) were created for each of the four OSF sub-tests.

For example, the first athlete achieved the score of 8.3 in 3 × 10 m run. Taking into account mean and standard deviation for all the athletes in this test (8.538 and 0.84, respectively), the normalised value after the correction came to approx. −0.285. Similarly, the remaining standardised values were as follows: 0.92; −1.367; −0.381. In this manner, a total value of −0.545 places this athlete in the 43rd percentile according to the chart of normalised values (Appendix A). It means that 43% of the children under study had poorer scores.

## 3. Results

The OSF test consists of four sub-tests assessing different motor abilities (speed, power, strength and endurance). A detailed description of the test can be found in the Methods section. The data presented in Table 2 show that endurance was a dominant ability, as the highest mean point value was obtained in 4-min run.

Table 3 shows the comparison of OSF test scores (points) in the years when research was carried out. A significant correlation was noted between all the analysed variables with regard to particular years (*p* ≤ 0.001). Small effect size was revealed, with the exception of standing broad jump (0.0642 points—medium effect size).

The distribution of results for each OSF sub-test is presented using colourful centile curves (Figure 1, Figure 2, Figure 3, Figure 4, Figure 5, Figure 6, Figure 7 and Figure 8). Centile is a value which shows the percentage of individuals who have achieved the same or worse results in comparison to a person with a given value of the explanatory variable. For example, if a person obtains a score equal to the 50th centile (yellow curve) for a given age group, it means that in a random sample of 100 same-age individuals, an average of 50 of them have lower scores. Centile values are also presented in Table 4, Table 5, Table 6, Table 7, Table 8, Table 9, Table 10 and Table 11 for particular OSF sub-tests taking into account sex.

**Figure 1 ijerph-19-14303-f001:**
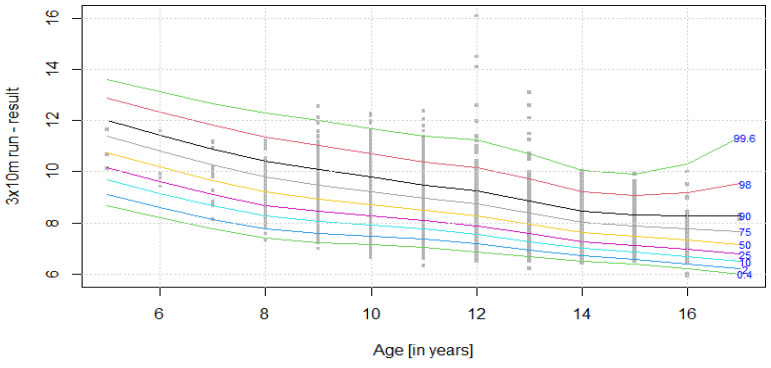
Centile chart for 3 × 10 m run in boys.

**Table 4 ijerph-19-14303-t004:** Values of particular centiles for 3 × 10 m run in boys.

Centile Age (Years)	0.4	2	10	25	50	75	90	98	99.6
**5**	8.68	9.10	9.68	10.17	10.76	11.39	12.01	12.88	13.62
**6**	8.19	8.59	9.14	9.62	10.19	10.81	11.44	12.34	13.13
**7**	7.76	8.13	8.66	9.11	9.65	10.27	10.89	11.82	12.68
**8**	7.41	7.77	8.26	8.69	9.21	9.80	10.42	11.37	12.29
**9**	7.23	7.58	8.04	8.44	8.93	9.48	10.08	11.03	12.00
**10**	7.16	7.49	7.92	8.28	8.72	9.23	9.79	10.69	11.67
**11**	7.03	7.35	7.75	8.09	8.50	8.97	9.49	10.37	11.38
**12**	6.87	7.17	7.56	7.88	8.27	8.73	9.24	10.15	11.24
**13**	6.66	6.92	7.27	7.57	7.94	8.38	8.87	9.72	10.72
**14**	6.48	6.70	7.00	7.27	7.61	8.02	8.47	9.23	10.05
**15**	6.38	6.58	6.87	7.13	7.46	7.87	8.32	9.09	9.92
**16**	6.21	6.40	6.69	6.96	7.32	7.76	8.28	9.20	10.30
**17**	5.99	6.19	6.49	6.77	7.16	7.67	8.29	9.54	11.36

**Figure 2 ijerph-19-14303-f002:**
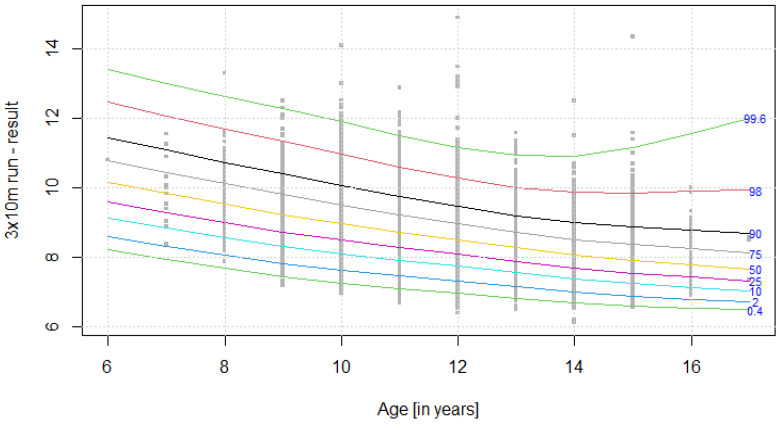
Centile chart for 3 × 10 m run in girls.

**Table 5 ijerph-19-14303-t005:** Values of particular centiles for 3 × 10 m run in girls.

Centile Age (Years)	0.4	2	10	25	50	75	90	98	99.6
**6**	8.21	8.60	9.13	9.60	10.16	10.80	11.46	12.46	13.4
**7**	7.95	8.33	8.85	9.29	9.84	10.45	11.09	12.06	13.0
**8**	7.69	8.07	8.57	9.00	9.53	10.12	10.73	11.69	12.6
**9**	7.46	7.83	8.32	8.74	9.24	9.81	10.40	11.34	12.3
**10**	7.26	7.62	8.10	8.50	8.97	9.51	10.07	10.98	11.9
**11**	7.11	7.46	7.92	8.29	8.73	9.23	9.75	10.60	11.5
**12**	6.97	7.32	7.75	8.09	8.50	8.97	9.46	10.28	11.2
**13**	6.82	7.15	7.56	7.88	8.27	8.72	9.20	10.01	10.9
**14**	6.69	6.99	7.38	7.69	8.07	8.51	9.00	9.87	10.9
**15**	6.60	6.88	7.25	7.55	7.92	8.37	8.89	9.86	11.2
**16**	6.55	6.80	7.14	7.43	7.80	8.25	8.79	9.90	11.6
**17**	6.50	6.73	7.04	7.32	7.68	8.13	8.70	9.94	12.0

**Figure 3 ijerph-19-14303-f003:**
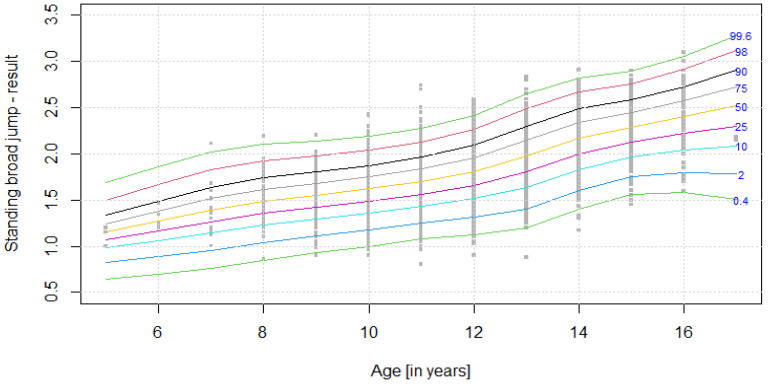
Centile chart for standing broad jump in boys.

**Table 6 ijerph-19-14303-t006:** Values of particular centiles for standing broad jump in boys.

Centile Age (Years)	0.4	2	10	25	50	75	90	98	99.6
**5**	0.647	0.827	0.98	1.07	1.16	1.24	1.33	1.49	1.69
**6**	0.698	0.888	1.06	1.17	1.27	1.38	1.49	1.66	1.86
**7**	0.756	0.953	1.14	1.27	1.39	1.51	1.64	1.82	2.02
**8**	0.843	1.034	1.23	1.35	1.49	1.61	1.74	1.92	2.10
**9**	0.930	1.107	1.29	1.42	1.55	1.68	1.80	1.98	2.13
**10**	1.000	1.172	1.36	1.49	1.62	1.75	1.87	2.04	2.19
**11**	1.077	1.246	1.43	1.57	1.70	1.84	1.96	2.13	2.27
**12**	1.128	1.311	1.51	1.66	1.81	1.96	2.09	2.27	2.41
**13**	1.197	1.406	1.64	1.81	1.98	2.15	2.30	2.49	2.65
**14**	1.398	1.602	1.83	2.00	2.17	2.34	2.48	2.67	2.82
**15**	1.563	1.752	1.97	2.12	2.29	2.45	2.58	2.76	2.90
**16**	1.585	1.800	2.04	2.22	2.40	2.57	2.72	2.91	3.06
**17**	1.505	1.782	2.09	2.30	2.52	2.73	2.90	3.12	3.29

**Figure 4 ijerph-19-14303-f004:**
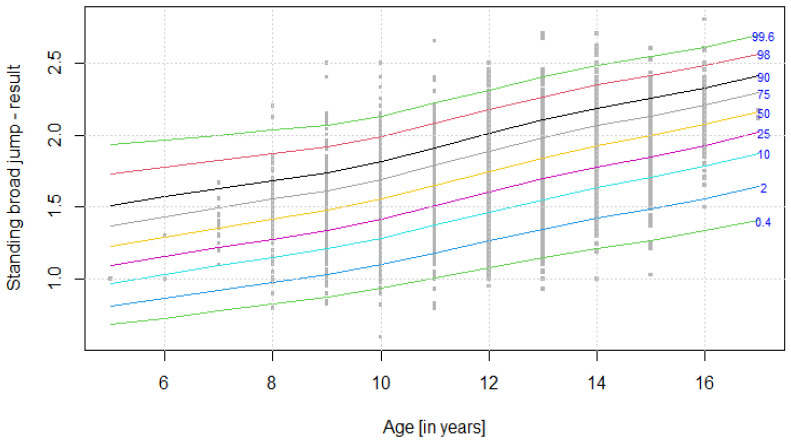
Centile chart for standing broad jump in girls.

**Table 7 ijerph-19-14303-t007:** Values of particular centiles for standing broad jump in girls.

Centile Age (Years)	0.4	2	10	25	50	75	90	98	99.6
**5**	0.686	0.816	0.973	1.09	1.23	1.37	1.51	1.73	1.93
**6**	0.732	0.869	1.032	1.16	1.29	1.43	1.57	1.77	1.96
**7**	0.779	0.923	1.092	1.22	1.35	1.50	1.63	1.82	2.00
**8**	0.827	0.977	1.151	1.28	1.42	1.55	1.69	1.87	2.03
**9**	0.876	1.032	1.211	1.34	1.48	1.61	1.74	1.92	2.07
**10**	0.937	1.100	1.284	1.42	1.55	1.69	1.82	1.99	2.13
**11**	1.009	1.183	1.375	1.51	1.65	1.79	1.91	2.08	2.22
**12**	1.079	1.265	1.465	1.60	1.75	1.88	2.01	2.17	2.31
**13**	1.148	1.345	1.554	1.70	1.84	1.98	2.10	2.27	2.40
**14**	1.213	1.422	1.637	1.78	1.93	2.06	2.19	2.35	2.48
**15**	1.270	1.488	1.707	1.85	2.00	2.13	2.25	2.41	2.54
**16**	1.335	1.561	1.785	1.93	2.07	2.21	2.33	2.48	2.61
**17**	1.408	1.644	1.872	2.02	2.16	2.29	2.41	2.56	2.69

**Figure 5 ijerph-19-14303-f005:**
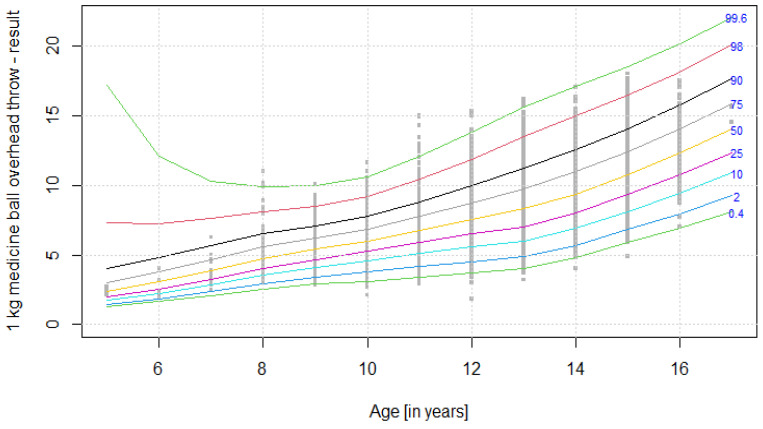
Centile chart for 1 kg medicine ball overhead throw in boys.

**Table 8 ijerph-19-14303-t008:** Values of particular centiles for 1 kg medicine ball overhead throw in boys.

Centile Age (Years)	0.4	2	10	25	50	75	90	98	99.6
**5**	1.32	1.47	1.72	1.98	2.38	3.02	4.04	7.32	17.17 *
**6**	1.65	1.86	2.20	2.54	3.04	3.77	4.78	7.27	12.08
**7**	2.08	2.37	2.83	3.26	3.87	4.66	5.65	7.62	10.26
**8**	2.55	2.95	3.53	4.05	4.73	5.55	6.49	8.11	9.93
**9**	2.90	3.41	4.10	4.68	5.40	6.21	7.08	8.49	9.94
**10**	3.11	3.76	4.58	5.24	6.00	6.85	7.74	9.16	10.62
**11**	3.37	4.17	5.15	5.91	6.79	7.76	8.78	10.41	12.13
**12**	3.68	4.51	5.60	6.49	7.55	8.73	9.96	11.88	13.80
**13**	4.03	4.85	6.01	7.03	8.29	9.73	11.21	13.46	15.58
**14**	4.82	5.66	6.89	7.99	9.38	10.96	12.58	14.98	17.13
**15**	5.90	6.80	8.13	9.31	10.77	12.42	14.08	16.45	18.51
**16**	6.93	7.96	9.44	10.73	12.30	14.04	15.74	18.12	20.14
**17**	8.07	9.24	10.91	12.33	14.04	15.88	17.64	20.06	22.06

* The score is affected by a small number of observations and wide confidence intervals. The application of additive models (GAMLSS) is one of the method components used when developing centile charts.

**Figure 6 ijerph-19-14303-f006:**
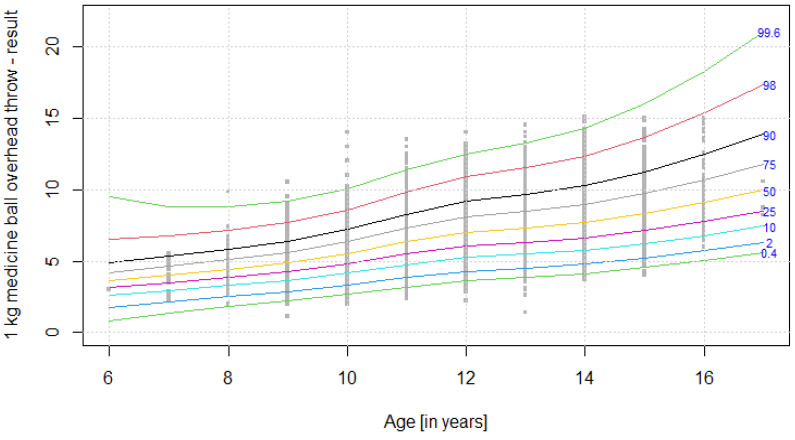
Centile chart for 1 kg medicine ball overhead throw in girls.

**Table 9 ijerph-19-14303-t009:** Values of particular centiles for 1 kg medicine ball overhead throw in girls.

Centile Age (Years)	0.4	2	10	25	50	75	90	98	99.6
**6**	0.835	1.74	2.63	3.15	3.66	4.22	4.90	6.53	9.52
**7**	1.359	2.15	2.96	3.49	4.04	4.64	5.33	6.74	8.82
**8**	1.819	2.52	3.29	3.83	4.43	5.09	5.81	7.13	8.80
**9**	2.232	2.88	3.65	4.23	4.89	5.61	6.38	7.70	9.18
**10**	2.680	3.33	4.14	4.78	5.53	6.35	7.21	8.60	10.06
**11**	3.183	3.86	4.75	5.48	6.34	7.30	8.29	9.83	11.36
**12**	3.595	4.29	5.24	6.03	7.00	8.07	9.18	10.88	12.52
**13**	3.846	4.53	5.48	6.31	7.33	8.49	9.70	11.54	13.29
**14**	4.122	4.79	5.75	6.61	7.69	8.95	10.28	12.33	14.30
**15**	4.533	5.21	6.21	7.12	8.31	9.72	11.24	13.63	15.97
**16**	5.042	5.74	6.80	7.79	9.10	10.70	12.47	15.35	18.25
**17**	5.592	6.32	7.44	8.51	9.96	11.78	13.85	17.35	21.04

**Figure 7 ijerph-19-14303-f007:**
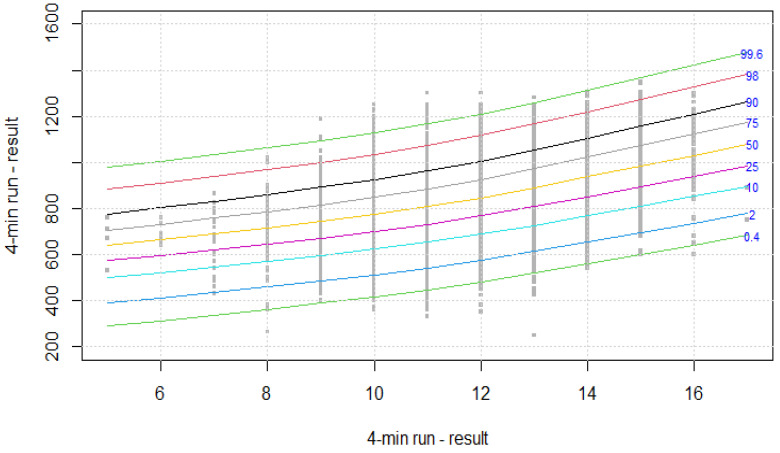
Centile chart for 4-min run in boys.

**Table 10 ijerph-19-14303-t010:** Values of particular centiles for 4-min run in boys.

Centile Age (Years)	0.4	2	10	25	50	75	90	98	99.6
**5**	290	390	501	573	639	704	775	883	979
**6**	313	412	523	596	663	730	803	911	1006
**7**	338	435	546	620	689	758	831	940	1034
**8**	363	460	570	645	716	787	861	970	1064
**9**	390	485	596	672	744	817	892	1002	1095
**10**	418	512	623	700	775	849	926	1035	1128
**11**	448	542	654	732	809	885	963	1074	1166
**12**	482	576	688	768	847	926	1006	1117	1210
**13**	519	613	726	808	890	973	1054	1167	1260
**14**	559	653	768	852	937	1022	1106	1220	1314
**15**	600	694	810	895	984	1072	1158	1273	1368
**16**	642	736	853	940	1031	1123	1210	1327	1421
**17**	685	779	897	986	1080	1175	1264	1382	1476

**Figure 8 ijerph-19-14303-f008:**
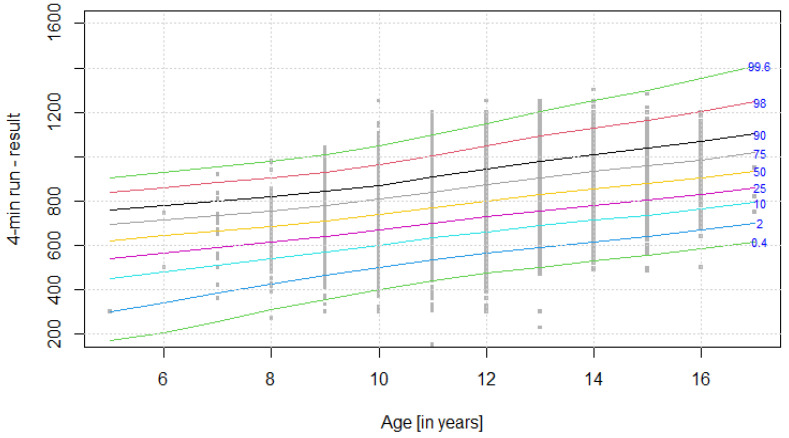
Centile chart for 4-min run in girls.

**Table 11 ijerph-19-14303-t011:** Values of particular centiles for 4-min run in girls.

Centile Age (Years)	0.4	2	10	25	50	75	90	98	99.6
**5**	171	304	451	540	622	696	759	840	906
**6**	208	343	480	564	643	715	778	861	931
**7**	257	385	511	589	665	736	799	883	955
**8**	309	426	540	614	687	756	819	905	981
**9**	359	464	570	640	711	779	842	931	1010
**10**	403	500	601	669	739	807	872	964	1048
**11**	442	535	633	701	771	841	908	1006	1098
**12**	475	565	663	730	802	875	945	1050	1151
**13**	502	591	688	757	830	905	979	1092	1204
**14**	530	617	713	781	855	932	1009	1130	1253
**15**	558	642	737	805	879	957	1037	1165	1299
**16**	587	671	764	832	906	986	1069	1204	1352
**17**	617	700	793	861	936	1017	1103	1247	1410

Significant high correlations were noted between all OSF sub-tests taking into account both scores and points (*p* ≤ 0.001)—Table 12.

Spearman’s correlation coefficient ranges from −1 to 1. Results of statistical analysis shown in Table 12 confirm the following correlations: the better the result in the test of speed, the higher the number of points; the longer the jump or the throw, the better the point score; the more metres covered in the test of endurance, the higher the point value. Moderate correlations were noted between the sum of points and the medicine ball throw (result and points) as well as between the sum of points and the 4-min run (result). In the case of other variables, strong correlations were observed. When it came to results of particular sub-tests, it was noted that individuals who had better scores in 3 × 10 m run (i.e., those who were faster) were also better at standing broad jump and 1 kg medicine ball throw (moderate correlation). An opposite correlation was found between 3 × 10 m run and the 4-min run. It shows that individuals with better 3 × 10 m run results obtained lower scores in the 4-min run (moderate correlation). What seems to be consistent with the specificity of the tests is that athletes who exhibit greater speed abilities achieve worse results in endurance tests and vice versa.

## 4. Discussion

The current study sought to validate a physical fitness test (OSF test) and to determine fitness norms for children and adolescents with a special focus on talent identification within its particular sub-tests of endurance, speed, strength and power. Based on the findings, centile charts were developed taking into account age and sex. The charts make it possible to interpret scores in particular motor sub-tests and, first and foremost, enable users to compare a given score with results obtained by their peers.

In part, physical fitness is genetically determined; however, to a large extent, it can also be shaped by environmental factors [37]. Unfortunately, children and adolescents rarely meet minimum recommendations concerning daily physical activity these days. What is more, their sedentary time increases gradually with age [38,39,40,41,42,43]. As a consequence, their PF levels decrease [44], and overweight and obesity become more prevalent [45,46].

Therefore, implementing PF tests in educational practice seems to be important on a population scale. Still, for optimal interpretation of PF levels in children, it is necessary to have current reference values coming from a random and largely representative sample of the studied population.

From the point of view of public health, education and sport, the development and implementation of PF tests seems to be justified [23,24,27,28,29]. Despite well-documented benefits stemming from higher PF levels in youth, there is an ongoing debate whether its levels in young people gradually decrease with age [47,48]. Chovanov et al. [49] and Pasichnyk et al. [50] reported a decreasing trend in PF levels and health status in the young generation. In England, PF levels have been decreasing by approx. 8% per decade—twice as fast as in other developed countries [51]. However, Moliner-Urdiales et al. [52] pointed to an increase in speed and agility but a decrease in strength in Spanish adolescents. As for Polish youth, normalised values of EUROFIT test decreased regularly in comparison with results from a previous decade (1989 vs. 1999 vs. 2009) [53].

The present study identifies and quantifies differences in OSF that are non-specific in terms of sex. Nonetheless, further research is needed to address these differences. There is a need to carry out cohort research to better understand what mechanisms contribute to sex- and age-specific differences in PF in childhood and adolescence. As stressed before, the aim of the current study was to normalise the OSF test. Now that the centile charts have been developed, the test may be widely used, and its results may be compared on a larger scale.

For practical reasons, PE teachers may play a crucial role in identifying children who demonstrate low levels of PF. Therefore, implementing PF tests in educational institutions seems to be an important public health issue. That is why close cooperation between educational bodies, health service and the government is indispensable [54].

Physical fitness tests may serve as an essential educational element, a tool for improving PF levels in children and as a monitoring system aimed at identifying children and adolescents with problems related to physical and motor development. They may help to address individual needs of children based on scientific evidence [55]. Owing to the fact that scores are converted into points taking into account age and sex, and because centile charts were developed as reference points, our OSF test seems to follow the aforementioned guidelines.

Ortega et al. [28] suggest using quintile-based normative frames to classify PF levels in children and adolescents, where individuals placed under 20th centile are classified as ‘very poor’, those between 20th and 40th centiles are considered to be ‘poor’, between 40th and 60th centiles‘medium’, between 60th and 80th centiles—‘good’, and those above 80th centile are classified as ‘very good’. This scale is also used when interpreting OSF test scores.

In their study, Dobosz et al. [56] revealed higher levels of PF in boys as well as increases in PF levels with age both in boys and in girls, which is also confirmed by our results.

The findings show sex- and age-specific percentile values for four sub-tests of OSF. Thus, it is possible to interpret and monitor the current state of PF in Polish children.

Colley et al. [57] analysed PF levels in individuals aged 6 to 19 over a 10-year period. They noted a decreasing trend in fitness in successive age groups. An important thing is that PF levels were higher in those who met PA recommendations. These observations are in line with our previous findings, which revealed significantly higher PF levels in OSF tests in children participating in AFA [58].

A proper interpretation of PF assessment requires comparing results obtained by an individual with normative values for the whole population in people of the same sex and age. Therefore, the current study presents normative values for specific age and sex and for the whole set of motor abilities in Polish youth. One advantage of the study is the strict standardisation when performing fitness tests in order to avoid errors resulting from inconsistency of measurement protocols. As a result, normative reference values for PF of Polish children and adolescents were developed. These values will ensure proper assessment and interpretation of PF levels in the young generation.

The 5th percentile achieved in the test may be used as a biological indicator, while values below this percentile may be treated as a pathological state [56,59]. From a practical standpoint, testing PF levels in this manner should take place in all schools in Poland. Physical education teachers should play a key role in identifying adolescents with low PF levels as well as highly talented individuals. Normative values should not be used to increase competition in children and adolescents. In turn, they should be applied to monitor progress made by individuals.

The findings of our study seem to be useful in terms of health promotion and sport, since they can be applied to identify those with too low PF levels and help to set adequate goals of health or sports training, monitor changes and promote health-oriented behaviours. According to Dobosz et al. [56], health promotion policy should be directed at promoting pro-health behaviours such as making sure proper PF levels are achieved and maintained since childhood.

Variability in physical fitness levels shown by our team does not include intra-individual differences between somatic features and motor test results. Moreover, it does not reflect changes stemming from ontogenesis of a given person, which was also noted by Dobosz [54]. We agree with this author’s statement, and we believe that using centile charts does make a lot of sense. If we do not deal with the issues of uneven development (allometry), the charts presented will be one of the few available tools for rational evaluation.

### Limitations

Reference values of physical fitness in children are best presented by longitudinal studies, since they make it possible to assess natural changes in the development of an individual. Our findings do not have such features.

The study participants were not tested in terms of the overall level of physical activity. Therefore, it is possible that their scores result not only from participating in the AFA programme but also from taking part in other activities that increase PF levels. On the other hand, the participants were not checked in terms of regularity of taking part in AFA training sessions. Moreover, the length of their training experience (duration of participation in the AFA programme) was not taken into consideration, either. All this may have affected their final scores in the fitness tests.

Socio-economic status of the study participants differed, so it is possible that the procedure of sample selection did not include all social classes. Our sex- and age-specific norms as well as differences in results were also limited by other unintended disturbing factors such as biological maturation. Given the above-mentioned limitations, we see the need to carry out research on the effects of maturation on physical fitness.

We reckon that the OSF test, owing to its simplicity as well as a possibility to assess motor abilities that are crucial from the point of view of athletics, is a valuable and innovative tool that we recommend should be commonly used in children and adolescents.

## 5. Conclusions

The centile charts developed by our team make it possible to compare a given result with results obtained by peers, to check if PF levels manifested are proper and to place scores achieved by an individual in an adequate centile channel.

The findings presented can be used in the process of sports training. In broadly understood sport, the result is the value itself. When comparing OSF test results with the whole population, we may identify sports talents and thereby rationalise the process of selection.

Finally, it is worth emphasising that in our work (creating centile charts for the OSF test) we aimed to develop reference points that would help to interpret results achieved in particular motor tests.

## Figures and Tables

**Table 1 ijerph-19-14303-t001:** Characteristics of the studied population.

Variable	Parameter	Total (N = 27,187)
Sex	Female	57.8% (N = 15,658)
	Male	42.2% (N = 11,416)
Age [in years]	N	27,187
	Mean (SD)	11.68 (1.59)
	Median (IQR)	12 (11–13)
	Range	5–17
Body height [cm]	N	25,634 *
	Mean (SD)	154.67 (11.55)
	Median (IQR)	154 (146–163)
	Range	108–198
Body mass [kg]	N	25,390 *
	Mean (SD)	43.49 (10.96)
	Median (IQR)	42 (35–50)
	Range	18–144
Year in which the study was conducted	2015	19.3% (N = 5246)
	2016	36.3% (N = 9866)
	2017	4.8% (N = 1292) **
	2018	39.7% (N = 10,783)

* N differs from N in total because not all study participants took part in body height and body mass measurements. ** sample size results from the decision of the Polish Athletic Association concerning obligatory participation of all AFA users in the International Physical Fitness Test in 2017. At that time, the OSF test was an additional (not obligatory) test.

**Table 2 ijerph-19-14303-t002:** OSF test scores (points).

Variable	Parameter	Total (N = 27,187)
3 × 10 m run	N	27,185 *
	Mean (SD)	61.96 (±15.72)
	Median (IQR)	65 (53–73)
	Range	0–100
Standing broad jump	N	27,186 *
	Mean (SD)	61.16 (±14.88)
	Median (IQR)	62 (54–70)
	Range	0–99
1 kg medicine ball	N	27,187 *
	Mean (SD)	57.95 (±14.8)
	Median (IQR)	60 (50–67)
	Range	0–99
4-min run	N	27,185 *
	Mean (SD)	63.81 (±15.29)
	Median (IQR)	66 (55–74)
	Range	0–98
Sum of points	N	27,187 *
	Mean (SD)	244.88 (±45.69)
	Median (IQR)	250 (218–275)
	Range	36–372

* N differs from N in total because not all study participants took part in each OSF sub-test.

**Table 3 ijerph-19-14303-t003:** OSF test scores (points) in particular years in which the study was conducted (Kruskal–Wallis test).

Variable	Parameter	2015 (N = 5246)	2016 (N = 9866)	2017 (N = 1292)	2018 (N = 10,783)	*p*-Value	Statistics	Effect Size
3 × 10 m run	N	5246	9866	1292	10,781	≤0.001	643.0463	0.0235
	Mean (SD)	64.38 (±11.83)	63.47 (±11.67)	48.98 (±20.86)	60.96 (±18.77)			
	Median (IQR)	67 (61–72)	65 (57–72)	48 (31–66)	63 (48–76)			
	Range	0–93	0–92	1–90	1–100			
Standing broad jump	N	5246	9866	1292	10,782	≤0.001	1747.7612	0.0642
	Mean (SD)	64.44 (±7.92)	59.22 (±9.31)	47.47 (±23.59)	62.99 (±18.64)			
	Median (IQR)	65 (62–69)	59 (53–65)	47 (30–65)	65 (50–79)			
	Range	0–91	0–86	1–94	0–99			
1 kg medicine ball throw	N	5246	9866	1292	10,783	≤0.001	458.3877	0.0168
	Mean (SD)	56.41 (±11.84)	58.07 (±11.05)	52.74 (±19.74)	59.21 (±17.88)			
	Median (IQR)	61 (53–64)	58 (50–65)	58.5 (37–67)	62 (50–70)			
	Range	0–83	0–91	0–95	1–99			
4-min run	N	5246	9866	1292	10,781	≤0.001	1099.1049	0.0403
	Mean (SD)	67.42 (±10.21)	63.72 (±11.6)	50.89 (±17.68)	63.7 (±18.77)			
	Median (IQR)	69 (64–73)	64 (56–72)	51 (36–64)	65 (50–80)			
	Range	0–91	0- 89	0–92	1–98			
Sum of points	N	5246	9866	1292	10,783	≤0.001	944.0356	0.0346
	Mean (SD)	252.65 (±31.34)	244.47 (±34.99)	200.09 (±64.41)	246.83 (±53.81)			
	Median (IQR)	258 (238–273)	246 (222–268)	200.5 (151.75–249)	252 (211–287)			
	Range	47–327	105–347	38–355	36–372			

**Table 12 ijerph-19-14303-t012:** OSF test results (raw scores and points)—Spearman’s correlation matrix.

	3 × 10 m Run—Points	Standing Broad Jump—Result	Standing Broad Jump—Points	1 kg Medicine Ball Throw—Result	1 kg Medicine Ball Throw—Points	4-Minute Run—result	4-Minute Run—Points	Sum of Points
**3 × 10 m run—result**	−0.822 ***	−0.700 **	−0.530 **	−0.477 **	−0.371 *	−0.576 **	−0.457 **	−0.720 ***
**3 × 10 m run—points**		0.476 **	0.587 **	0.226 *	0.298 *	0.422 **	0.473 **	0.784 ***
**Standing broad jump—result**			0.784 ***	0.590 **	0.461 **	0.585 **	0.443 **	0.705 ***
**Standing broad jump—points**				0.355 *	0.414 **	0.428 **	0.487 **	0.810 ***
**1 kg medicine ball throw—result**					0.875 ***	0.409 **	0.248 *	0.536 **
**1 kg medicine ball throw—points**						0.313 *	0.264 *	0.624 **
**4-min run—result**							0.889 ***	0.676 **
**4-min run—points**								0.733 ***

* weak correlation. ** moderate correlation. *** strong correlation.

## Data Availability

Not applicable.

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
