# Peer review of "Assessment of Motor Abilities and Physical Fitness in Youth in the Context of Talent Identification—OSF Test"

_ijerph, 2022, doi:10.3390/ijerph192114303_

Round 1
Reviewer 1 Report
Thank you for the opportunity to review this interesting manuscript. In general, studies that address the health of young people in the community are important, as well as the development of instruments/strategies for health assessment. Here are my considerations:
Introduction:
- I consider it good, however, I suggest expanding and deepening the information:
1. Exemplify the motor skills that are part of a PF (i.e., strength, resistance, flexibility, balance, etc.);
2. In the Discussion section (lines 294-296), the authors wrote "Unfortunately, children and adolescents rarely meet minimum recommendations concerning daily physical activity these days". Therefore, it would be very important to include information on the level of physical inactivity of young people in the Introduction section, in addition to international recommendations on the time of weekly physical activity for this population;
3. Still in the Discussion section, the authors address the topic of obesity, this could/should already be indicated in the Introduction section;
3. In the paragraph before the presentation of the objectives of the study, it is essential that the authors improve the Justification for carrying out this study. Why carry out this study? What is missing from current literature? What would be the gaps at the moment that justify the publication of the study?
Methods:
1. I suggest an excellent explanation of the reasons that led the authors not to include abdominal strength and flexibility tests in this battery of tests. Detail why you chose these 4 tests (only these motor skills);
2. Statistics: considering the large and excellent number of study participants, it would be very important to inform that before the analysis the data were tested for normality. And if they were, what test was used!
3. Include information about correlations in this section;
3. every study has confounding factors! The authors did not control for this in the analyses, nor did they justify the fact in the limitations. I suggest reviewing this point...
Results:
1. I suggest presenting the values ​​used for the interpretation of correlation coefficients in the Methodology section and not below Table 12. In addition to adding the reference for these values!
Discussion:
1. I suggest starting this section by detailing the main findings of the study. I suggest that the first paragraph be restricted to clear and accurate information about what was achieved. So far, the Discussion consists of a more in-depth Introduction;
2. The information presented in lines 320-325 would be a great strategy for the Introduction and not the discussion of the results;
*Overall, this section should be completely rewritten. I justify my request to the fact that the authors never "directly" confront their results with the previous literature. They only present results from other studies: please review this;
**I suggest reviewing the method/technique used by other studies and following this as an example.
Limitations:
In my view, there are still limitations to be justified:
1. Why other motor skills were not included in this battery, such as balance, abdominal strength, flexibility;
2. The study may be biased, as many children participated in training groups, others may not. And this was not controlled in the analyses...
Minor considerations:
1. Disparity in spaces between lines from section 2. Materials and Methods (line 54);
2. In several passages sentences start outside the tabulation required by the journal (i.e., line 63).
Author Response
Dear Reviewer,
Thank you for your important suggestions - we have tried to incorporate them into the text. We hope that the article in this form will be accepted.
Please see the attachment.

Reviewer 2 Report
Thank you for the opportunity to read the interesting work, however, the authors should complete it. To supplement the keywords with the quality of life. In the introduction, they should include the concept of health-related quality of life, which includes physical fitness and cite Chmielik LP, Mielnik-Niedzielska G, Kasprzyk A, Stankiewicz T, Niedzielski A. Physical and Psychosocial Concept Domains Related to Health-Related Quality of Life ( HRQL) in 50 Girls and 52 Boys Between 5 and 18 Years Old in Poland Using the Parent-Reported 50-Item Child Health Questionnaire (CHQ-PF50). Med Sci Monit. 2022 Jun 9; 28: e936801. doi: 10.12659 / MSM.936801. and Lin, X.J.; Lin, I.M.; Fan, S.Y. Methodological issues in measuring health-related quality of life. Tzu Chi Med. J. 2013, 25, 8–12.
[CrossRef] In the material and the method, the criteria for selection for the study group are not clear. They should also be given the criteria for admission and exclusion in the study group. The text of the work should include information about the consent of the bioethics committee
Author Response

(The authors gave the same response as above.)

Reviewer 3 Report
Dear Authors,
First, I want to emphasize the relevance of the research and the huge amount of work done. The obtained results are exciting and really necessary for teachers, and coaches, especially in the context of a constant decrease in the volume of physical activity, and accordingly, the level of physical fitness of children and youth. I am convinced that the normative values will be useful not only at the level of Poland but also interesting for researchers from other countries. at the same time, I want to express a few of my remarks, which will be useful for a clearer structure of the article and emphasize its relevance.
1. The introductory part should be more meaningful, at the moment it does not reflect the current state of the problem, does not emphasize the relevance of the study, does not provide information about the features and necessity of physical fitness testing in Poland, etc. In my opinion, some of the method information is more relevant to the introductory part (lines 55-80)
2. The materials and methods section should be structured more. It is worth separating a separate section related to the organization and conduct of the research (currently, this part of the material is in the section related to the description of the sample). Information on bioethical expertise and related issues can be a separate subsection or in the description of the research organization (currently, this material is found in the description of the sample and in the description of the statistical analysis).
It is also worth noting the criteria for the inclusion of participants in the study
3. It is not entirely clear what the year of study (year of conducting the research?) means in table 1.
4. Given the fact that children of different genders and very different age ranges are involved in the study, what is the meaning of presenting the raw test scores in Table 2? Since the values of these tests will differ in different age groups
5. The topic of the article emphasizes the identification of talents, while neither the discussion nor the conclusions provide an answer to this question. Also, in the discussion, it is worth providing more information that would indicate the practical significance of the obtained results. It is worth further analyzing the result given in Table 12 and its significance in connection with the stated research topic
Kind regards,
Author Response

(The authors gave the same response as above.)

Round 2
Reviewer 1 Report
Dear, I consider that after adjustments the manuscript has improved a lot in every way! Congratulations... I have doubts about the presentation of the results. Due to the large number of images and scarce text between the Figures, there is a problem: The reader is lost, unable to connect the whole. Therefore, I suggest "still" adopting some strategy to make this section lighter. In the present form, there are many images (all important), but this makes it difficult to understand the Results. Accept this as a suggestion and technical detail, which does not prevent the publication of the text.
Author Response
Dear Reviewer,
Thank you for your important suggestions. Please see the attachment.
Yours sincerely
Authors

Reviewer 2 Report
I would like to thank the authors for the information sent and for making partial corrections. Unfortunately, the work still requires some more corrections.
1- in the introduction you should find the concept of quality of life, especially health-related quality of life, which links physical activity with other extremely important aspects of human life and therefore the literature should be supplemented with Chmielik LP, Mielnik-Niedzielska G, Kasprzyk A, Stankiewicz T, Niedzielski A. Physical and Psychosocial Concept Domains Related to Health-Related Quality of Life (HRQL) in 50 Girls and 52 Boys Between 5 and 18 Years Old in Poland Using the Parent-Reported 50-Item Child Health Questionnaire (CHQ-PF50 ). Med Sci Monit. 2022 Jun 9; 28: e936801. doi: 10.12659 / MSM.936801. and Lin, X.J .; Lin, I.M .; Fan, S.Y. Methodological issues in measuring health-related quality of life. Tzu Chi Med. J. 2013, 25, 8-12.
2- the selection criteria for the study group are still not clearly stated (are there only 27187 children in Warsaw and Biała Podlaska who all train athletics ?; are all children training athletics in Warsaw and Biała Podlaska included in The 'Athletics for All' (AFA) ) if not, which of them and in what way were included in this ptogram)
3-criteria for inclusion and exclusion from work are a very important part of the work, meanwhile authorizations still specify the inclusion criteria are too vague the authors write "Membership of the AFA group was also an inclusion criterion (age, participation in athletics training, written parental consent" criteria of exclusion do not say at all, whether I understand that the age of the participants was 0-100 years ?.
If only the athletics group was tested, the results cannot be related to the general population and be a criterion for assessing it, as the authors suggest later in the paper, but they can only be applied to athletes practitioners.
4- "Mean age of the participants was 11.68 (± 1.59) years and it was slightly lower than median age (12 years). Girls constituted a larger proportion of the population under study (57.8%) - see table 1. "should be transferred to the results
Author Response

(The authors gave the same response as above.)

Reviewer 3 Report
Dear Authors,
Thank you for all clarifications.
Kind regards,
Author Response
Dear Reviewer,
Thank you for your important suggestions and acceptance of our article.
Yours sincerely
Authors